# Mixture Distributions for Scalable Bayesian Inference

## Abstract

Bayesian Neural Networks (BNNs) provides a mathematically grounded framework to quantify uncertainty. However BNNs are computationally inefficient, thus are generally not employed on complicated machine learning tasks. Deep Ensembles were introduced as a Bootstrap inspired frequentist approach to the community, as an alternative to BNN's. Ensembles of deterministic and stochastic networks are a good uncertainty estimator in various applications (Although, they are criticized for not being Bayesian). We show Ensembles of deterministic and stochastic Neural Networks can indeed be cast as an approximate Bayesian inference. Deep Ensembles have another weakness of having high space complexity, we provide an alternative to it by modifying the original Bayes by Backprop (BBB) algorithm to learn more general concrete mixture distributions over weights. We show our methods and its variants can give better uncertainty estimates at a significantly lower parametric overhead than Deep Ensembles. We validate our hypothesis through experiments like non-linear regression, predictive uncertainty estimation, detecting adversarial images and exploration-exploitation trade-off in reinforcement learning.

## 1 Introduction

Neural Networks models have been applied in diverse fields from weather forecasting, to autonomous vehicle driving, to online advertisement and many more (Goodfellow et al. (2016)). However, vanilla feed-forward NNs are susceptible to the problem of over-fitting. In addition to this, NNs trained using Maximum Likelihood Estimation (MLE), or Maximum A Posteriori (MAP) cannot provide an estimate of the uncertainty in predicted value and tend to produce overconfident results on *out-of-distribution* test data. Thus we aim to build deep learning frameworks which are more robust, secure and reliable - specifically, in the face of uncertainty we would like the model to be able to say *"I Don't Know!"*

A principled approach to build such models is through Bayesian inference, where instead of a point estimate for the network parameters, we infer the *posterior* distribution of the weights given the data. The predictive distribution of unseen data $(x^*, y^*)$ is given by $q(y^*|x^*) = \int p(y^*|x^*, W)p(W|D)dW$, where $p(W|D)$ is the true posterior computed from Bayes rule. Unfortunately, since modern NNs have an exponentially large number of parameters and do not lend themselves to being integrated into mathematical equations, exact inference remains intractable.

Several methods for approximate Bayesian inference have been investigated. Until now, Hamiltonian Monte Carlo (HMC) (Neal (2012)) has been considered the gold standard for approximate Bayesian inference. However, HMC requires explicit storage of samples from the posterior and is typically not scalable to larger networks and datasets. Thus methods based on Variational Inference (VI) have gained popularity recently for the task of approximate inference. VI relies on using a surrogate posterior $(q_\theta(W))$ as an approximation for the true posterior. One of the first application of VI in NNs was by Hinton & Van Camp (1993) but the optimization remained intractable for most Bayesian Neural Networks (BNNs). Graves (2011) revisited similar ideas and proposed a simple but biased estimator for performing VI with a fully factorized posterior (mean field assumption).

Recently, *Bayes-by-Backprop* algorithm (BBB) also employed VI using an unbiased estimator for Variational loss, a fully factorized Gaussian posterior and a non-Gaussian prior Blundell et al. (2015). Interestingly, Gal & Ghahramani (2016) showed links between Bernoulli Dropout(Srivastava et al. (2014), Hinton et al. (2012)) and approximate inference in Deep Gaussian Process (Damianou & Lawrence (2013)), thus allowing for the extraction of uncertainty estimates of a NN in a principled way. Louizos & Welling (2016) arrived at the same conclusion through structured posterior approximations via matrix variate Gaussian Posterior (Gupta & Nagar (2018)) and local reparameterization trick (Kingma et al. (2015)).

Bayesian methods are far more computationally expensive, as compared to standard NNs, making Deep Ensembles a very attractive alternative as they scale well to large datasets and models. The intuition behind Deep Ensembles is straightforward, NNs initialized and trained independently can be expected to output similar predictions on training data, while disagreeing on points away from training set. Despite their empirical success in uncertainty estimation and adversarial robustness (Lakshminarayanan et al. (2017), Strauss et al. (2018)), Deep Ensembles have lacked theoretic support from the Bayesian perspective. Thus, it is interesting to see how ensembles of deterministic and stochastic NNs can be placed into the VI framework. Another drawback of Deep Ensembles is their heavy parametric overhead. We modify the BBB algorithm to learn mixture distributions rather than the unimodal Gaussian distribution, which reduces this burden. We further show that this way, Deep Ensembles can be linked with the Bayesian framework. In essence, using our methods, we are able to achieve accuracy and uncertainty estimates (with lower parametric cost) which are never worse than Deep Ensembles, BBB, Concrete Dropout

**Contributions:** We design an efficient and simple way of sampling from mixture distributions which incorporates the *"local reparameterization trick"*(Kingma et al. (2015)) to obtain low variance gradients for tuning of parameters in mixture distributions. This is also of independent interest for designing more complex posteriors using any mixture distributions besides the ones used in this paper. We also fit ensembles of deterministic and Bayesian NNs into VI framework with no changes made to the original algorithm. We improve upon the uncertainty estimates of Deep Ensembles by using mixture distribution with relatively less number of parameters.

**Distribution of the paper** - In section 2, we show how the original BBB algorithm can be modified for generalized mixture distributions while remaining back-prop compatible. For simplicity, we stick to a mixture of Gaussian's. In section 3, we provide links between Ensembles and VI with mixture distributions. Section 4 contains experiments that demonstrate that our methods generate superior uncertainty estimates as compared to unimodal distributions or Deep Ensembles.

## 2 CONCRETE MIXTURE OF GAUSSIAN

A feed-forward network trained with gradient descent will arrive at point estimates. However, in the case of BNNs, the weights are not point estimates but a probability distribution. Let us revisit the problem of finding a functional relationship $y = f^W(x)$ between input x and output y, given a labelled training dataset D = $\{(x_1, y_1), (x_2, y_2), \dots\}$ from a Bayesian standpoint. Our task is to find a distribution over the parameters $p(W|D)$, given the input data. Given this posterior, we can predict test output $y^*$ given a new test input $x^*$ by marginalizing the likelihood over the space of parameters $W$.

However, even for modest sized NNs, the number of uncertain parameters prohibit this calculation analytically. We require *approximate inference* methods in such cases. We define an approximating *variational* distribution $q_\theta(W)$ with parameters $\theta$. We would minimize the Kullback-Leibler (KL) Divergence with respect to the parameters $\theta$ to maximize the similarity between proposed posterior and the true posterior. Note that the problem of minimizing KL Divergence can be reformulated to minimizing of another term usually called the Variational free energy in literature Friston et al. (2007), Blundell et al. (2015). Evaluating the objective of maximizing this is computationally expensive for large datasets therefore. Thus we split the entire data into M mini batches $D^1, D^2, \dots, D^M$. The unbiased Monte Carlo (MC) approximation to the minibatch cost it given as:

$$W^j \sim q_\theta(W), \quad \hat{\mathcal{L}}_{VI-MC}(\theta) = -\sum_{i \in D^j} \log(p(y_i|f^{W^j}(x_i))) + \frac{1}{M} KL(q_\theta(W)||p(W)) \quad (1)$$

In order to use stochastic backpropagation, we need the MC sample $W^j$ to be reparameterized as $g(\theta, \epsilon)$. We shall see how this is possible for Mixture of Gaussians (MoG) next.

### 2.1 SAMPLING FROM CONCRETE MIXTURE OF GAUSSIAN

Blundell et al. (2015) proposed the *Bayes-by-Backprop* algorithm for tuning of weight posterior's parameters. For simplicity, the posterior was assumed to be fully factorized Gaussian. The assumption of fully factorized posterior is also known as the Mean Field assumption in the literature.

$$q_\theta(W) = \prod_{i=1}^N \mathcal{N}(w_i; \mu_i, \sigma_i^2) \qquad \theta = (\mu_i, \sigma_i^2)_{(i=1,\dots,N)} \qquad N = \text{Total no. of weights} \quad (2)$$

Louizos & Welling (2016) used a matrix variate Gaussian posterior Gupta & Nagar (2018) to capture correlations between weights of same layer. Both of the above assumption do not account for multiple modes in true posterior. A natural choice for posterior that can capture multiple modes is

a mixture distribution, where each mixture component would correspond to a separate mode of the true posterior.

We show method to sample from for mixture of Gaussian (MoG) for independent weights and mixture of matrix variate Gaussian (MoMG) for correlated weights, while still being able to use the local reparameterization trick. This method can be extended for any mixture of distributions in general.

To sample weight $w$ from a mixture of K Gaussians i.e. $w \sim q_\theta(w) = \sum_{j=1}^{K} p_j \mathcal{N}(w; \mu_j, \sigma_j^2)$, $\sum_{j=1}^{K} p_j = 1$, we assume a latent random variable which governs the component of the mixture that generates the value of $w$. We assume the latent random variable to be uniform random variable u i.e. $u \sim \text{Uniform}[0,1]$. The latent random variable u has a $p_j$ probability of being in the range $[p_{j-1}, p_j + p_{j-1})$, similarly it has $p_{j+1}$ probability of being in between $[p_j + p_{j-1}, p_j + p_{j-1} + p_{j+1})$ and so on, thus if $u$ falls in the range $[p_{j-1}, p_j + p_{j-1})$ we sample from the j'th distribution $\mathcal{N}(w; \mu_j, \sigma_j^2)$. For example when sampling from mixture of three Gaussian, we will have to first sample two random variables $z_{p_1}$ and $z_{p_1+p_2}$, which can be seen as correlated Bernoulli random variables.

$$u \sim \text{Uniform}[0,1], \quad z_{p_1} = \mathbb{1}_{u < p_1}, \quad z_{p_1+p_2} = \mathbb{1}_{u < p_1 + p_2} \quad and \quad \epsilon \sim \mathcal{N}(0,1)$$

$$w = z_{p_1}(\mu_1 + \sigma_1 \epsilon) + (1 - z_{p_1})(z_{p_1+p_2})(\mu_2 + \sigma_2 \epsilon) + (1 - z_{p_1+p_2})(\mu_3 + \sigma_3 \epsilon) \quad (3)$$

Unfortunately, in the above case where we sample $w$ from a multimodal distribution, local reparametrization trick cannot be applied. This is because the Bernoulli random variables $z_{p_1}$ and $z_{p_1+p_2}$ can't be parameterized in the form of $g(\theta, \epsilon)$. So, we replace them with their continuous relaxation, the Concrete distribution Gal et al. (2017),Maddison et al. (2016). This distribution can be viewed as softmax relaxation of the max function used in Gumbel-max trick Gal et al. (2017). This allows us to represent the discrete random variable $z_{p_1}$ as $\hat{z}_{p_1} = g(\theta, \epsilon)$. Thus we replace $z_{p_1}$ and $z_{p_1+p_2}$ in equation 4 with:

$$z_{p_1} \approx \hat{z}_{p_1} = \text{sigmoid}(\frac{1}{t}.(\log(p_1) - \log(1 - p_1) + \log(u) - \log(1 - u))$$

$$z_{p_1+p_2} \approx \hat{z}_{p_1+p_2} = \text{sigmoid}(\frac{1}{t}.(\log(p_1 + p_2) - \log(1 - p_1 - p_2) + \log(u) - \log(1 - u)) \quad (4)$$

where, t is the temperature term and is usually kept as 0.1 or 0.01 ($<<1$ to guarantee most of the mass of the pdf is around 0 and 1). So, we have now successfully reparameterized $w$ as $g(\theta, \epsilon)$ where $\theta = (\mu_1, \sigma_1, \mu_2, \sigma_2, \mu_3, \sigma_3, p_1, p_2)$. To prevent parametric explosion, we assume $p_1$, $p_2$ are same for the entire layer. This is similar to assuming that Dropout parameter p is the same for entire layer. Thus now, with mean field assumption we can now define the posterior of the weight matrix $W$ from mixture of K Gaussians as:

$$q_\theta(W) = \prod_i \sum_j p_{ij} \mathcal{N}(w_i; \mu_{ij}, \sigma_{ij}^2), \quad \sum_{j=1}^{K} p_{ij} = 1, \quad \theta = ((p_{ij}, \mu_{ij}, \sigma_{ij}^2)_{j=1,...,K})_{i=1,...,N} \quad (5)$$

To model the correlations among the weights it is possible for us the treat the entire weight matrix $W$ of a layer, as a MoMG Gupta & Nagar (2018) i.e. $p(W) = \sum_i p_i \mathcal{MN}(M_i, U_i, V_i)$, where

$$p(W') = \mathcal{MN}(M_i, U_i, V_i) = \frac{\exp(-\frac{1}{2}tr[V_i^{-1}(W' - M_i)^T U_i^{-1}(W' - M_i)])}{(2\pi)^{rc/2}|V_i|^{c/2}|U_i|^{r/2}} \quad (6)$$

$M_i$ is a r $\times$ c matrix and is the mean of the distribution, $U_i$ is a r $\times$ r matrix that provides covariance of the rows and $V_i$ is a c $\times$ c matrix that governs the covariance of the columns of the matrix. It has been shown by Gupta et al. in Gupta & Nagar (2018) that matrix variate distributions are essentially multivariate Gaussian.

$$p(\text{vec}(W')) = \mathcal{N}(\text{vec}(M_i), V_i \otimes U_i), \ \otimes \text{ is the Kronecker product} \quad (7)$$

Thus, now instead of treating each weight independently we are now able to sample directly from its joint distribution which is assumed to be a MoMG. For sampling from a mixture of 2 matrix variate

Gaussian, $W = z_p(M_1 + U_1^{\frac{1}{2}} E V_1^{\frac{1}{2}}) + (1 - z_p)(M_2 + U_2^{\frac{1}{2}} E V_2^{\frac{1}{2}}))$ where $E_{i,j} \sim \mathcal{N}(0,1)$. We replace $z_p$ with its continuous relaxed version $\hat{z}_p$, thus making it possible to use the local reparameterization trick. We assume $U_i = \text{diag}(\sigma_{r_i}^2)$ and $V_i = \text{diag}(\sigma_{c_i}^2)$ i.e. independent rows and columns in this paper. This allows us to reduce the number of parameters as compared to MoG, while still modelling correlations between weights. Now that we are able to sample from MoG and MoMG we need to be able to approximate $D_{KL}(q_\theta(W)||p(W))$, where we assume a Gaussian prior i.e. $p(W) = \mathcal{N}(0, l^{-2}I)$. We use Monte Carlo approximations similar to the one used in Blundell et al. (2015) for optimising the $D_{KL}(q_\theta(W)||p(W))$.

For all following sections, we interchangeably use the name Concrete MoG (CMoG) and Concrete MoMG (CMoMG) for MoG and MoMG respectively.

## 3 DEEP ENSEMBLE AS BAYESIAN INFERENCE

Deep Ensemble is a simple non-Bayesian framework which is able to predict uncertainty on *out-of-distribution* samples. Deep Ensemble algorithm involves training **M** randomly initialized networks on independently sampled batches. Owing to its simplicity and scalability, Deep Ensemble is widely used for uncertainty estimation. This departure from Bayesian methodology to capture uncertainty is of concern since the Bayesian framework is a principled and widely accepted approach to capture uncertainty. Thus, it is indeed surprising to see that with no simplification made to the original Deep Ensemble algorithm described in Lakshminarayanan et al. (2017), we can show equivalence with the VI framework.

In order to show the equivalence with VI, we assume the approximate posterior is a mixture of points or Gaussian (with $\sigma \to 0$) in a high dimensional probability space. For example in case the network has only two layers with parameters $W_1$ and $W_2$ respectively, the joint distribution is $q(W_1, W_2|\theta) = p\mathcal{N}(M_1, \sigma^2 I) + (1 - p)\mathcal{N}(M_2, \sigma^2 I)(\sigma \to 0)$, where $\theta$ is the set of all variational parameters which in this case are $M_1$ and $M_2$. We fix probability $p = \frac{1}{2}$.

Note, $M_1 = [M_{11}, M_{12}]$ is the concatenation of the variational parameters of layer 1 and layer 2, and similarly, $M_2 = [M_{21}, M_{22}]$ is the concatenation of the variational parameters of layer 1 and layer 2 (Assuming we have a NN of only two layers for simplicity). For example if the first point is selected while sampling, we will have $W_1 = M_{11}$ and $W_2 = M_{12}$, we will see that minimizing the KL divergence of this posterior with the true posterior is mathematically equivalent to the Deep Ensemble algorithm described above. For sampling, the $W_1$ and $W_2$ from the above posterior, we use the same sampling technique as described for MoG, except that here same random variable $z_p$ is tied to all variational parameters.

$$u \sim \text{Uniform}[0, 1] \quad \text{and} \quad z_p = \mathbb{1}_{u < p}$$
$$W_1 = z_p M_{11} + (1 - z_p)M_{21}, \qquad W_2 = z_p M_{12} + (1 - z_p)M_{22} \qquad (8)$$

In MoG for every weight we had to sample a separate $z_p$ independently for each weight. This random variable tying is the only difference in sampling between the two, and this observation is a critical tool linking Deep Ensemble to VI Framework. Next we try to minimize the unbiased Monte Carlo (MC) estimate of the mini-batch variational free energy. Minimizing the mini-batch cost for mini-batch i = 1, 2 ... M

$$w^{(i)} \sim q_\theta(w), \quad L_{VI-MC}^i = -\log(p(D^{(i)}|w^{(i)})) + \frac{1}{M}\log(q(w^{(i)}|\theta)) - \frac{1}{M}\log(p(w^{(i)})) \quad (9)$$

Note we have taken a MC estimate of the KL divergence too, but following Hoffman et al. (2013),Kingma & Welling (2014) convergence to the same limit as the expectation of variational free energy is still guaranteed. Now assume a Gaussian prior i.e. $P(w^{(i)}) = \mathcal{N}(0, l^{-2}I)$. Note for $p = \frac{1}{2}$ there are equal chance of having $w^{(i)} = M_1$ or $w^{(i)} = M_2$, suppose for this batch i, $w^{(i)} = M_1$

$$L_{VI-MC}^i = -\log(p(D^{(i)}|M_1)) + \frac{1}{M}\log(\frac{1}{2}) + \frac{1}{M}\frac{1}{2}l^2||M_1||_2^2 \qquad (10)$$

We further simplify loss function for regression case, and fit a Gaussian with variance $\tau^{-1}$ (also called model precision), to the negative log-likelihood of the data i.e. $p(y_j|x_j, M_1) = \mathcal{N}(y_j; \hat{y}_j, \tau^{-1}I)$. So, after dividing the whole equation by $\tau$ we have

$$L_{VI-MC}^i \propto \frac{1}{2}\sum_{j \in D^i} ||y_j - \hat{y}_j||_2^2 + \frac{1}{M\tau}\log(\frac{1}{2}) + \frac{1}{M}\frac{1}{2\tau}l^2||M_1||_2^2 \qquad (11)$$

Minimizing the above equation can be seen as training a single NN with parameters $M_1$ on mini batch i with L2 regularization $\lambda = \frac{1}{2M\tau}l^2$. Equivalently, there will be another mini-batch j where $w^{(j)} = M_2$, and minimizing the MC estimate of the variational free energy, in this case, will be equivalent to training a NN with parameters $M_2$ with the same regularization as before. This is equivalent to a Deep Ensemble of 2 NNs with parameters $M_1$ and $M_2$ which were randomly initialized and trained on independently sampled mini-batches as was in the original algorithm. The predictive distribution of unknown label $y^*$ is given as $q(y^*|x^*) = \int p(y^*|x^*,w)q(W|\theta)dW = \frac{1}{2}p(y^*|x^*, M_1) + \frac{1}{2}p(y^*|x^*, M_2)$, which is a mixture of Gaussian and matches the predictive distribution of Deep Ensemble. Thus independently training an Ensemble of NN's can be seen as training a "hypothetical" Bayesian NN's with a specific mixture distribution.

Although we have shown equivalence for a specific case of 2 NNs, it can be easily generalized to N NNs, by considering an approximate posterior a mixture of N points and taking $p = \frac{1}{N}$. Note that we can tune $p$'s too by using the same concrete relaxation that we took in CMoG. We called this modified algorithm Concrete Ensemble.

Stein Variational Gradient Descent (SVGD) (Liu & Wang (2016)) is particle based Bayesian approach and is close to Deep Ensembles. Both SVGD and Deep Ensembles can be viewed as particle filters trying to approximate an underlying distribution. The mini-batch SVGD update is given as $\theta_i^{t+1} = \theta_i^t + \frac{1}{N}\sum_{j=1}^N [k(\theta_j^t, \theta_i^t)\nabla_{\theta_j^t}\log(p(\theta_j^t|D^i)) + \nabla_{\theta_j^t}k(\theta_j^t, \theta_i^t)]$. Ensembles have a mode seeking nature, whereas SVGD seek to maintain some level of diversity. SVGD is far less popular despite their similarity with Deep Ensembles because in practice, different initializations for large scale NNs suffice for capturing different modes, and thus, the additional time complexity overhead of $O(N^2)$ for computing the kernel is not justified by empirical results.

More specifically, if $f_\theta(w) = \frac{1}{N}\sum_{i=1}^N f_i(w; \theta_i)$, where $\theta_i$ are independent variational parameters, and $f_i(w)$ could be any arbitrary distribution. Let us define $L(\theta)$ as the Variational Free Energy of the mixture distribution $f_\theta(w)$, and $L(\theta_i)$ as the Variational Free Energy of the individual mixture component $f_i(w; \theta_i)$. We can approximately bound $L(\theta)$ as; $\frac{1}{N}\sum_{i=1}^N L(\theta_i) - H(\frac{1}{N}) \le L(\theta) \le \frac{1}{N}\sum_{i=1}^N L(\theta_i)$, where H(p ) is the entropy. So, individually optimizing $\theta_i$'s (since they are independent of each other) is also minimizing the variational free energy of the mixture distribution, and interestingly increasing the ensemble size increases the tightness of the bound.The upper bound is arrived at using the convexity of KL divergence and the lower bound is obtained by further lower bounding the variational approximation to KL divergence between mixture distributions (Durrieu et al. (2012)). See Appendix A.2 for more details

Simple arguments that will allow us to extend this proof to ensemble of arbitrary architecture of Neural Networks have been added to Appendix 3.

## 4 EXPERIMENTS

The following networks are used in experiments: Normal feed forward Multi Layer Perceptron (MLP), Deep Ensemble of 3 NN (Ensemble)(Lakshminarayanan et al. (2017)), Concrete Deep Ensemble of 3 NN (CEnsemble), Concrete Mixture of 3 Gaussain (CMoG) with $p$'s tied for each layer (3 $p$'s per layer), Concrete Mixture of 2 Matrix Variate Gaussian (CMoMG), Bayes by Backprop (BBB) (Blundell et al. (2015)) and Concrete Dropout (CDropout) (Gal et al. (2017)).

### 4.1 REGRESSION TASK

We use Concrete mixture of Gaussian, and Concrete Ensemble described above to fit $y = x^3$ for $-4 < x < 4$, and observe the estimation for $-6 < x < 6$. We observe our algorithms (CMoG and CMoMG) can predict uncertainty in out of training data points much better compared to the uncertainty estimation with vanilla NN or previous uncertainty estimation frameworks like MC-Dropouts Gal (2016), Ensemble, BBB, and Concrete dropouts. For both CMoG (we consider a mixture of 2 and 3 Gaussian) and Concrete Ensemble, we use fully connected feed forward NN with two hidden layers of size 100 each and ReLU activation, and performed a forward pass ten times, with result averaging for training, and for testing, performing a forward pass on trained NN forty times. We perform this experiment thrice and average the results. Figure 1 compares uncertainty estimation using CMoG and Concrete Ensemble with previous frameworks for the same architecture as described above.

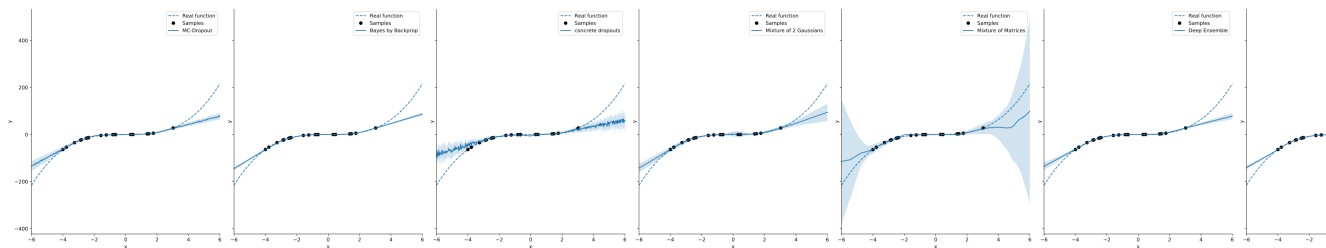

Figure 1: Uncertainty estimates ($\pm 1$ std ) for out of distribution samples on toy regression data

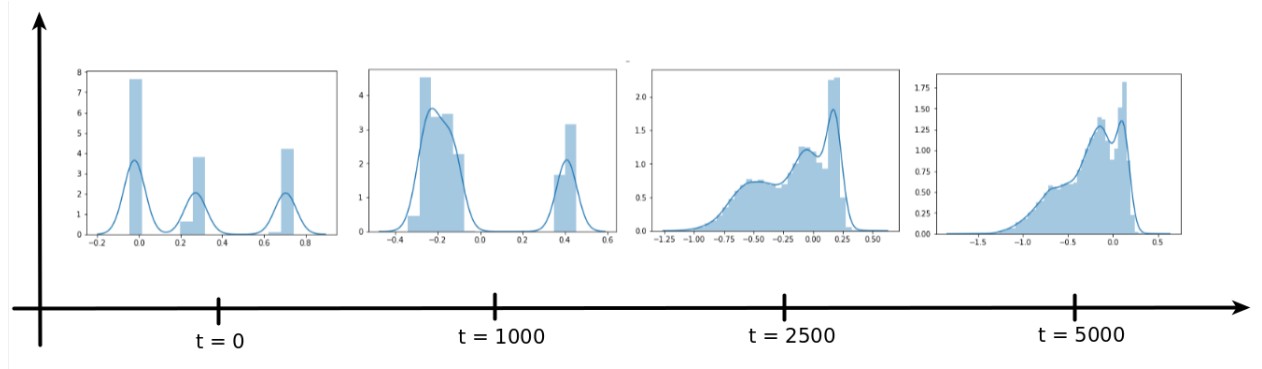

Figure 2: Time evolution of weight distribution of one of the weight in the CMoG-BNN with mixture of 3 Gaussians

## 4.2 UNCERTAINTY ESTIMATION ON FULLY CONNECTED NETWORKS

In this subsection, we show advantage of using multimodal posterior for fully connected feed forward networks. We first describe results for our methods on the classification task of MNIST digit dataset (LeCun et al. (2010)). We use a fully connected network with two hidden layers having 1000 and 600 units respectively and ReLU activation. The last layer is a softmax layer with ten units. Refer appendix section for accuracy obtained and no. of parameters used.

Now, we assess how our methods behave when test data does not lie on the training manifold. Overconfident predictions on such data are one of the most important challenges for reliable deployment of deep learning methods in the real world. The expectation is that if the input image does not lie on the training manifold, then the predictive uncertainty should be high. We use the same architecture as described above for the remaining subsection.

The first experiment we do is evaluate the predictive uncertainty of model trained on MNIST dataset on Fashion MNIST dataset (Xiao et al. (2017)). This is similar to MNIST-NotMNIST experiment done by Lakshminarayanan et al. (2017). As the images in Fashion MNIST dataset are completely different than in MNIST, ideally a trained model should be very uncertain about its prediction. Figure 3 shows the histogram of entropy of the output of trained models using different methods on test MNIST, and test Fashion MNIST data. MLP and Concrete Dropout are overconfident on predictions for Fashion MNIST data. Deep Ensemble and CEnsemble show an increase in no. of data points showing high entropy. BBB shows significant improvement over previously described methods. CMoMG shows comparable results with BBB even after having more modes indicating that the assumption of independent rows and columns may not be best for capturing uncertainty. CMoG gives the best results by far, which are also near ideal expectations.

For second and third experiment we use more sophisticated methods for analyzing behavior of predictive uncertainty of model as done in Li & Gal (2017). We don't include MLP and Concrete Dropout for these experiments for better clarity (they perform significantly worse). The second experiment considers single gradient step Fast Gradient Sign (FGS) method (Goodfellow et al. (2014)). The method generates an adversarial input by adding small perturbations to an input image by moving in the direction which reduces predicted class probability.

$$\mathbf{x}_{adv} = \mathbf{x} - \eta \, \mathrm{sgn}(\nabla_{\mathbf{x}} \max_{y} \log p(y|\mathbf{x})) \qquad (12)$$

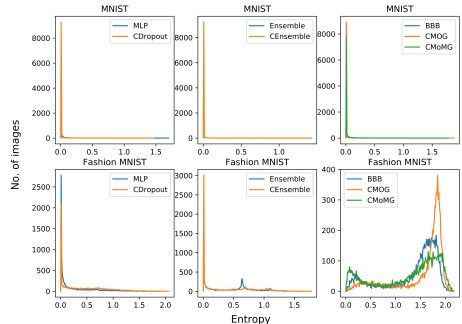

Figure 3: MNIST-FashionMNIST : Histogram (200 bins) of entropy obtained on MNIST and Fashion MNIST test data

The step size $\eta$ is varied from 0 to 0.5. Figure 4 shows the accuracy and predictive entropy for different values of $\eta$. CMoG again gives the highest predictive entropy which helps in identifying adversarial inputs. CMoMG has the slowest decaying accuracy vs step-size curve which makes it more robust to adversarial attacks. For example, second column of images($\eta = 0.13$) in right panel of Figure 4 are visually close to true class '2' and CMoMG gives a higher accuracy for that $\eta$ than any other method. Ensemble based methods are again outperformed by BBB, CMoG and CMoMG. The second attack we consider is the targeted version of FGS (Goodfellow et al. (2014)). The attack

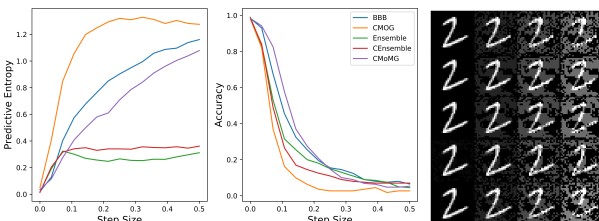

Figure 4: Untargeted Attack: Classification accuracy and predictive entropy vs step-size. The adversarial images are shown(top to bottom) for BBB, CEnsemble, Ensemble, CMoG and CMoMG

here is implemented iteratively (Kurakin et al. (2016)) which produces adversarial examples by even finer perturbations than single gradient step and without completely destroying the content of original image. The predictive probability of a target class ('0' in our case) is maximized for all non-target class images using

$$\mathbf{x}_{adv}^0 = \mathbf{x}, \qquad \mathbf{x}_{adv}^{t+1} = \mathbf{x}_{adv}^t + \eta \, \mathrm{sgn}(\nabla_{\mathbf{x}_{adv}^t} \log p(y_{target}|\mathbf{x}_{adv}^t)) \tag{13}$$

We fix $\eta$ to 0.01 and iteratively update for 100 iterations. The results obtained are shown in Figure 5. CMoMG has the slowest decaying and slowest rising true class and target class accuracy respectively making it more robust against targeted attack. CMoG has highest rise in predictive entropy, making it more suited for identifying adversarial attack. We note that for iteration 30-100, CMoMG has better ability to identify an adversarial input. Both these methods having multi-modal posterior perform better than BBB and Ensemble based methods.

More detailed analysis is required for examining the performance of our methods on adversarial attacks, but we conclude from the experiments that multi-modal posterior based methods are well suited for capturing uncertainty.

### 4.3 UNCERTAINTY ESTIMATION ON ALEXNET AND RESNET

We implement multimodal posterior for AlexNet (Krizhevsky et al. (2012)) and ResNet (He et al. (2016))for getting better uncertainty estimates. The posterior used is only for fully connected layers before final layer thus, with less no. of parameters than Deep Ensemble. All the other layers except fully connected layer have been used with spatial concrete dropout. Refer appendix section for accuracy obtained and no. of parameters used.

We repeat the MNIST-FashionMNIST experiment done in subsection 4.2 for AlexNet. Figure 6(L) shows the result obtained. We have shown CDF for better comparison. Note that, even with slightly less no. of parameters CMoG provides considerable improvement over Deep Ensembles while BBB

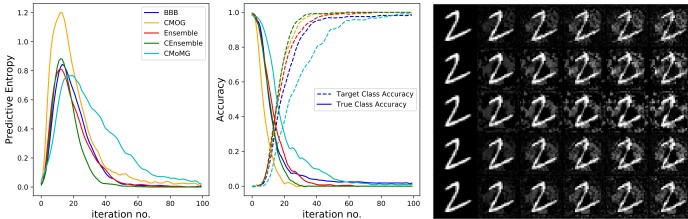

Figure 5: Targeted Attack: Classification accuracy for target class '0' and true class and predictive entropy vs. iterations. The adversarial images are shown(top to bottom) for BBB, CEnsemble, Ensemble, CMoG, and CMoMG

and CMoMG gives comparable uncertainty with significantly less no. of parameters. This effect will only improve for larger and deeper networks.

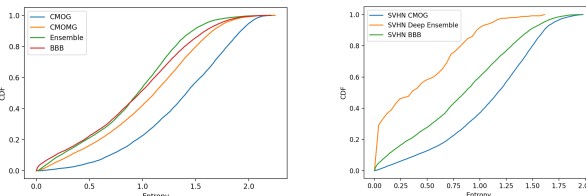

Figure 6: Predictive Entropy CDF: AlexNet on FashionMNIST (L), ResNet on SVHN (R)

We repeat the same experiment but now for CIFAR-10 (Krizhevsky et al.) - SVHN (Netzer et al. (2011)), using ResNet-20. Figure 6(R) shows the uncertainty obtained on SVHN dataset after training the model on CIFAR-10 dataset. A clear improvement on uncertainty estimation can be observed for CMoG over Deep Ensembles with significantly less no. of parameters. This further proves that we can indeed get better uncertainty estimates for deep networks by using multimodal posterior on few layers with only a small increase in parameters over normal network.

### 4.4 CONTEXTUAL BANDIT

We consider an online learning task (Mushroom Bandit Task Collier & Llorens (2018), Blundell et al. (2015)) to get a clear handle on the usefulness of uncertainty estimates. In an online setup, the agent plays her action based on a policy $\pi$. Based on this action, the environment reveals a stochastic reward. The objective of online learning is to Update the policy $\pi$ as more and more data is revealed to maximize the cumulative reward received. Since the reward revealed is a noisy reward, online learning algorithms face a fundamental dilemma: exploration v/s exploitation. Typically, online learning algorithms explore in high uncertainty environments and exploit in low uncertainty setups.

We train a contextual neural bandit to learn the dependency between the context and optimal action as per the pseudo-algorithm given in Collier & Llorens (2018). We train the NN based on BBB, CMoG, Ensemble, CEnsemble, and CMoMG. The regret variation is shown in figure 7 with the results have been averaged over three trials.

The MoG agent achieves a balance between eating and ignoring mushrooms more quickly than an ensemble network and incurs the least regret. Similarly, Matrix Variate based CMoMG can estimate uncertainty better than BBB and learns the context much faster. Thus, we can see that the ability to capture multiple modes provides us superior estimates of uncertainty, allowing us to balance exploration-exploitation in a better way.

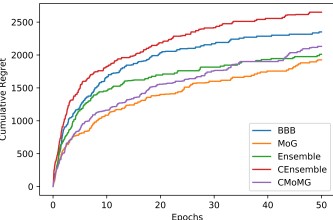

Figure 7: Cumulative Regret Variation for Mushroom Bandit Task

## 5 DISCUSSION

In this paper, we show equivalence between Ensembles of Deterministic and Bayesian NN and VI. We have also generalized this proof to an ensemble of NN's with arbitrary architecture. Bayesian NN's with multi-modal posterior have been shown to give superior uncertainty estimates on out of distribution samples, be more robust to adversarial attacks and learn better trade-offs between exploration and exploitation, while maintaining comparable or better test accuracy as standard methods. We also show that the parametric overhead of Ensembles can be reduced by using Bayesian NN's with mixture distribution.

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

# A  APPENDIX

## A.1  ACCURACY AND NUMBER OF PARAMETERS

Table 1 reports the test accuracy obtained on MNIST with Fully connected network and results were averaged over three trials. The accuracy of CMoG , CEnsemble and CMoMG are better or comparable to benchmark MLP, BBB, Deep Ensemble and Concrete Dropout with the same architecture.

Table 1: Test Accuracy on MNIST

| Method | Test Accuracy | No. of parameters (in Millions) |
|---|---|---|
| MLP | 98.28 | 1.39 |
| Concrete Dropout | 98.8 | 1.39 |
| Deep Ensemble | 98.51 | 4.17 |
| BBB | 98.44 | 2.78 |
| Concrete Ensemble | 98.57 | 4.17 |
| CMoG | 98.6 | 8.34 |
| CMoMG | 98.66 | 2.79 |

Table 2 reports the test accuracy obtained on MNIST with AlexNet and results were averaged over three trials.

Table 2: Test Accuracy on MNIST

| Method (Dataset) | Test Accuracy | No. of parameters (in Thousands) |
|---|---|---|
| CMoG | 99.36 | 44.94 |
| BBB | 99.26 | 21.58 |
| Deep Ensemble | 99.29 | 48.25 |
| CMoMG | 99.18 | 29.38 |

Table 3 reports the test accuracy obtained on CIFAR-10 with Resnet-20 and results were averaged over three trials.

Table 3: Test Accuracy on CIFAR-10

| Method (Dataset) | Test Accuracy | No. of parameters (in Millions) |
|---|---|---|
| CMoG | 91.9 | 0.273 |
| BBB | 91.01 | 0.271 |
| Deep Ensemble | 92.1 | 0.81 |

Table 4 reports the Expected Calibration Error (ECE) on CIFAR-10 with Resnet-20 and results were averaged over three trials.

Table 4: ECE on CIFAR-10

| Method | ECE |
|---|---|
| CMoG | 0.019 |
| MoG (categorical p) | 0.023 |
| Deep Ensemble | 0.018 |

## A.2 BOUNDING THE VARIATIONAL FREE ENERGY WITH RESPECT TO MIXTURE DISTRIBUTIONS

Variational free energy has two components the Expected loss and the KL divergence between the assumed posterior and the prior. Now because the Expectation over a mixture distribution is equal to the sum over expectation of individual components all we need to do is bound the KL divergence between mixture distribution and true posterior in terms of KL divergence between individual mixture components and the true posterior.

Thus we want to bound the KL divergence between f(x) and g(x) where

$$f_\theta(x) = \sum_{i=1}^{N} p_i f_i(x; \theta_i)$$

$p_i$ is the weight of the component individual i in the mixture distribution.

A simple upper bound can be achieved using convexity of KL distance as

$$D_{KL}(f_\theta(x)||g(x)) \leq \sum_{i=1}^{N} p_i D_{KL}(f_i(x; \theta_i)||g(x)) \tag{14}$$

Keeping $p_i$ fixed as $\frac{1}{N}$ will give us the desired upper bound

We can also find the lower bound using the Variational Approximation (Hershey & Olsen (2007)) by first splitting the KL divergence into two components-

$$D_{KL}(f(x)||g(x)) = E_{f(x)}[\log(f(x))] - E_{f(x)}[\log(g(x))] \tag{15}$$

We can use Jensen's inequality to get the appropriate lower bound for the first term as-

$$
\begin{aligned}
E_f(\log(f(x))) &= \sum_i p_i \int_x f_i(x) \log(\sum_j p_j \frac{\phi_{ji}}{\phi_{ji}} f_j) dx \\
&\geq \sum_{ij} p_i \phi_{ji} \int_x f_i(x) \log(p_j f_j(x)/\phi_{ji})
\end{aligned}
$$

Maximizing the lower bound wrt $\phi_{ji}$ under the constraint $\phi_{ji} \geq 0$ and $\sum_j \phi_{ji} = 1, \forall$ i,j we get

$$E_f(\log(f(x))) \geq \sum_{i=1}^{N} p_i \log(\sum_{j=1}^{N} p_j e^{-D_{KL}(f_i||f_j)}) - \sum_{i=1}^{N} p_i H(f_i)$$

By similar arguments we can derive -

$$E_f(\log(g(x))) \geq \sum_{i=1}^{N} p_i \log(e^{-D_{KL}(f_i||g)}) - \sum_{i=1}^{N} p_i H(f_i)$$

Subtracting the two gives us the Variational Approximation (Hershey & Olsen (2007)) to KL divergence

$$D_{var}(f||g) = \sum_{i=1}^{N} p_i D_{KL}(f_i||g) + \sum_{i=1}^{N} p_i \log(\sum_{j=1}^{N} p_j e^{-D_{KL}(f_i||f_j)})$$

Durrieu et al. (2012) empirically showed this approximation to be closer to $D_{KL}(f||g)$ than most other approximations, thus $D_{var}(f||g) \approx D_{KL}(f||g)$

We further lower bound, this Variational approximation by ignoring all $j \neq i$ terms inside $\log(\sum_{j=1}^{L} p_j e^{-D_{KL}(f_i||f_j)})$

$$D_{KL}(f||g) \approx D_{var}(f||g) \geq \sum_{i=1}^{N} p_i D_{KL}(f_i||g) + \sum_{i=1}^{N} p_i \log(p_i)$$

Fixing $p_i$'s as $\frac{1}{N}$ gives us the required lower bound

$$D_{KL}(f||g) \approx D_{var}(f||g) \geq \frac{1}{N} \sum_{i=1}^{N} D_{KL}(f_i||g) - H(\frac{1}{N})$$

We know Variational free Energy w.r.t $f_\theta(w)$ is $L(\theta) = \sum_{i=1}^{N} E_{f_i(w)}(-\log(p(Y|X,w))\frac{1}{N} + D_{KL}(f_\theta(w)||p(w))$

We then use the derived upper and lower bounds for the KL divergence between mixture distribution to bound the Variational free energy as required.

### A.3 Generalization of Proof to be Model Agnostic

In an ensemble it is often the case when we take an ensemble of arbitrary NN's rather than NN's with same architecture. So it is desirable for a generalization of the proof given in Section 3 to this case.

To do so we add "pseudo parameters" to each NN such that the dimensionality of the parameters in each NN is extended to become the same and we assume this to be N. We can now define a prior on the $\mathcal{R}^N$ such that the support of both the approximate posterior Mixture of Gaussian and Gaussian prior is the same.

We call these added parameters "pseudo" because we want them to be hanging parameters that do not effect the neural network output (imagine an additional hidden layer whose output is detached from the next layer, thus increasing the dimensionality of the network without changing it's original output).

Thus using this trick we have reduced the problem of ensemble of arbitrary NN's to the problem of NN's with same parametric dimensionality. Now we can define a mixture distribution posterior as $q(w|\theta) = \sum_{i=1}^{N} p_i q(w|\theta_i)$ where $\theta_i$ corresponds to the variational parameters of $i^{\text{th}}$ NN and each $\theta_i$ now has the same $\mathcal{R}^N$ dimensionality thanks to the previously mentioned trick.

Note now $\theta$ will comprise of two sets of variational parameter $\theta^r$ (real parameters) and $\theta^p$ (pseudo parameters). An interesting observation that can be made is that the optimal solution for the $^p$ is collapsing to the prior.

$$L(\theta) = E_{q(w|\theta)}[-\log(p(D|w)] + D_{KL}(q(w|\theta)||p(w))$$

Now for optimal solution w.r.t to $\theta^p$ we have -

$$\nabla_{\theta^p} E_{q(w|\theta)}[-\log(p(D|w)] + \nabla_{\theta^p} D_{KL}(q(w|\theta)||p(w)) = 0$$

Note $\nabla_{\theta^p} E_{q(w|\theta)}[-\log(p(D|w)] = 0$ by design, so clearly the optimal solution for $\theta^p$ is to collapse to the prior, e.g. if $q(w|\theta^p)$ is Gaussian and the prior $p(w) = \mathcal{N}(0, I)$ then we have $q(w|\theta^p) = \mathcal{N}(0, I)$ as the optimal solution. For the $q(w|\theta^r)$ the optimization shall remain exactly the same as in Section 3. Implying ensembles of NN's with different architecture too can be framed inside a Variational Framework.

