# OpenReview forum: "Mixture Distributions for Scalable Bayesian Inference"
_ICLR.cc/2020/Conference — Reject_

### Official Review · AnonReviewer3 · 2019-10-23
**Official Blind Review #3**

**Rating:** 1

**Review:**

Summary: This paper proposes to use either relaxed mixture distributions or relaxed mixtures of matrix Gaussians as the approximate posterior for Bayesian neural networks. Naturally, taking the mixture variance to zero allows a stretched interpretation of ensembling to become Bayesian as well. Experiments are performed on small neural networks on regression tasks, as well as uncertainty estimation experiments on deep networks. Finally, a downstream task of bandit optimization (the mushroom experiment) is performed.

post-rebuttal: I'd like to sincerely thank the authors of the paper for their enlightening discussions about their work. I'm inclined to think that this paper has a future, and definitely encourage the authors to resubmit - taking into account the reviews here. In particular, it would be greatly helpful if they cleaned up some of the writing and explanation, as our discussion points to.

However, my stance on the codebase remains mostly unchanged - while a notebook ended up being released with their method, it was too close to the deadline and in a bit too rough of shape to really be able to poke through their implementation.
From a quick pass, their implementation seems to be correct - although rather than attempting to base their code on the old torch/lua implemenations, I'd still suggest using a pre-written version. However, without the completely trained numbers, I cannot in good conscience vote to accept the paper.

Tldr: I vote to reject this paper for several reasons. The interpretation of ensembling as variational inference is both flawed and well known. Mixture distributions have been previously proposed for variational inference. There are significant enough weaknesses in the empirical results that make me question the authors’ own implementations. I will increase my score if these issues are resolved.

Originality:
1)	It is well known (and immediately obvious) that one can interpret ensembles as a Bayesian method – with a mixture of delta functions around the independently trained models. Furthermore, the Bayesian bootstrap (Rubin, 1983) is a Bayesian interpretation of the bootstrap (identical under many conditions), while re-weighted ensembles are Bayesian models (Newton, 1995; Newton, 2018) that converge to the true posterior under many conditions – intriguingly, this version of re-weighting could potentially have good uncertainty quantification properties. For the specific case of stacking (which is closely related to ensembling), there are interpretations of stacking as Bayesian model combination (Yao et al, 2018, Clyde & Iversen, 2013). As a result, this view of ensembling as a Bayesian method is not novel.

2)	Additionally, the derivation of dropout as Bayesian inference (which relies on a similar interpretation on taking the variance parameter in the Gaussian to zero) is known to be flawed (Hron et al, 2018); this flaw is directly applicable to the interpretation of ensembling as variational inference as described in Section 3 of this paper. The issue is fundamental: when taking the variance parameter to zero, the supports of the variational distribution and the true distributions have a mismatch, causing the KL divergence to become infinite. As a result, one cannot minimize the objective (as it is infinite by design).

Clarity:
1)	The proposed methods are clearly explained in the setting under which one might be able to re-implement them, particularly in the description of the approximate posteriors.

post rebuttal: to my knowledge, this was never really addressed. In particular, "why do we want to interpet ensembling in a Bayesian manner?" is still an open question to me.
2)	Why is that we should expect (approximate) Bayesian procedures to be better calibrated than frequentist ones? It may be the case that if the integrals are accurately estimated (e.g. we have reached the true posterior and the model used for the data is in fact accurate), then we should expect Bayesian procedures to be well calibrated. However, this does not seem to be the case. A reference or argument for why (even in the model incorrect & approximate inference cases) in the introduction would greatly enhance the strength of the paper.

post rebuttal: I think that this issue was addressed in a reasonable manner through the rebuttal, but will require a bit of rewriting.
3)	From a quick pass through Appendix A.2, the derivation is quite unstructured and difficult to follow. It is tough to see exactly what is being meant by the probabilities in the first line.
Additionally, the phrasing of “We further lower bound…” seems to suggest that the bound being used is not the well known bound of Durrieu et al, 2012, but a looser bound. If so, what is the approximation accuracy of this bound in relation to the other bounds compared against in this setting?

4)	In sum, three different methods are proposed: concrete mixtures of Gaussians, concrete ensembles, and concrete mixtures of matrix variate Gaussians. Throughout the experiments, it seems like __one__ of these methods always wins on the task at hand. However, it does not always seem to be the same method. Is there a recommendation for practitioners as to which method to use – or is this simply task dependent? Specifically, is there an understanding as to why each method performs vastly differently on each task?
A priori, I would expect the concrete mixture of matrix variate Gaussians to always perform best because it does seem to be able to model multivariate posteriors the best. However, it seems to perform worse on several of the predictive uncertainty tasks.

5)	For the uncertainty experiments, it would be nice to have more qualitative numbers – for example the expected calibration error – rather than just the plots on in and out of distribution example. After all, not only do we want to be able to recognize out of distribution examples, but we want to be able to trust the probabilities that are output.

6)	In Figure 4, thank you for providing the adversarial attacks used to fool the networks – the step size parameter doesn’t give a great intuition, and the images are welcome and useful.


Significance:
post rebuttal: I think that this issue was satisfactorily addressed.
1)	Mixture distributions as the variational family have already been proposed – at least once – see Miller et al, 2017 for an example that additionally uses the reparameterization trick. There, updates to the posterior are performed via selectively adding mixture components when necessary. Given that they also run experiments on neural networks, it would be nice to see a comparison to that method.

Quality:
1)	I am very concerned by the performance of the ResNet20 models trained on CIFAR10 in the paper. The reported results for deep ensembles (an ensemble of _three_ independently trained models) in Appendix A are 85.23% accuracy. By comparison, the number in the original paper (He et al, 2015) is 91.25% for a _single_ model. Furthermore, the first link to a PyTorch repo from a Google search (https://github.com/akamaster/pytorch_resnet_cifar10) comes up with a re-implementation that gets 91.63% (again for a single model).

__This issue alone is enough for me to vote to reject the paper until the authors’ implementations are fixed and run with deep ensembled ResNets that get similar accuracy. I hope to see the implementation issues fixed in the rebuttal period. __

2)	Furthermore, in Appendix 2, I see a potential issue in using the KL divergence and would like the authors to clarify.
If the secondary lower bound is being used to approximate the KL divergence between the mixture distributions, it becomes very unclear if what is being optimized is in fact a __lower bound__ on the log probability. After all, the standard ELBO is written as:
	Log p(y) \geq ELBO := E_q(\log p(y | \theta)) – KL(q(\theta) || p(\theta))
Implying that another lower bound on the KL would be greater than the ELBO and thus not necessarily a lower bound on the log marginal likelihood.

3)	Bottom of page 7: “The posterior is used only for fully connected layers…” Why is this the case? Alternative Bayesian methods can be used on the full network.


Minor Comments:
-	Please attempt to use the math_commands.tex in the ICLR style file to write math. This makes the math standardized throughout.
-	Figure 1 is especially unclear and the legends and captions should be made considerably larger. Zooming in at 800% seems to be necessary to even be able to make out which method is which. Consider, placing all methods on a single plot and then coloring/dashing them separately. Additionally, please compare to HMC on this task (and an equivalent RBF GP) to show the uncertainties from the “gold standard” methods.
-	Please additionally make the figure legends larger for the rest of the figures.
-	What is the meaning of time in Figure 2? I assume that it means training time, but it is not especially clear from the caption. With that being said, I think that it is quite interesting that the marginals do seem to look multi-modal, even at the end of training.
-	Line before Eq 7: please use \citet if possible to cite Gupta & Nagar, rather than stating the clunky Gupta et al. \citep{Gupta & Nagar}.
-	Eq 2: please do not use \hspace{-…} to save space in equations, so that the equation does not overwrite lines before.
-	Page 2: “Deep Ensembles have lacked theoretical support …” In order to make this claim, you must first explain why one ought to be Bayesian in the first place. See above for the history of interpreting ensembling from a Bayesian perspective. There is no a priori reason why one would not just wish to have frequentist justifications of ensembling (and potentially even calibration).
-	Please do not capitalize Ensembling or Variational Inference throughout.

References:
Clyde & Iversen, Bayesian model averaging in the M-open framework. In Bayesian Theory and Applications, 2013. DOI:10.1093/acprof:oso/9780199695607.003.0024

He, Zhang, Ren & Sun, Deep Residual Learning for Image Recognition, CVPR, 2016. https://arxiv.org/abs/1512.03385

Hron, Matthews & Ghahramani, Variational Bayesian dropout: pitfalls and fixes, ICML, 2018. https://arxiv.org/abs/1807.01969

Miller, Foti & Adams, Variational Boosting: Iteratively Refining Posterior Approximations, ICML, 2017. https://arxiv.org/abs/1611.06585

Newton & Raftery, Approximate Bayesian Inference with the Weighted Likelihood Bootstrap. JRSS:B, 1994. https://www.jstor.org/stable/2346025?seq=1#metadata_info_tab_contents

Newton, Polson & Xu, Weighted Bayesian Bootstrap for Scalable Bayes, 2018; https://arxiv.org/abs/1803.04559

Rubin, The Bayesian Bootstrap, 1981. Annals of Statistics. https://projecteuclid.org/euclid.aos/1176345338

Yao, Vehtari, Simpson, & Gelman, Using Stacking to Average Bayesian Predictive Distributions, Bayesian Analysis, 2018; https://projecteuclid.org/euclid.ba/1516093227


**Experience Assessment:**

I have published one or two papers in this area.

**Review Assessment: Checking Correctness Of Derivations And Theory:**

I carefully checked the derivations and theory.

**Review Assessment: Checking Correctness Of Experiments:**

I carefully checked the experiments.

**Review Assessment: Thoroughness In Paper Reading:**

I read the paper thoroughly.

---

> ### Author Response · Authors · 2019-11-14
> **Thank You for the Detailed Review**
>
> “The interpretation of ensembling as variational inference is both flawed and well known”,”Additionally, the derivation of dropout as Bayesian inference (which relies on a similar interpretation on taking the variance parameter in the Gaussian to zero) is known to be flawed (Hron et al, 2018); this flaw is directly applicable to the interpretation of ensembling as variational inference as described in Section 3 of this paper. The issue is fundamental: when taking the variance parameter to zero, the supports of the variational distribution and the true distributions have a mismatch, causing the KL divergence to become infinite. As a result, one cannot minimize the objective (as it is infinite by design).
>
> Indeed, if we take the posterior as a mixture of impulses then due to pathological KL divergences optimisation is impossible. Thus, we use a trick of keeping sigma sufficiently small. Gal had used this same trick for showing Dropouts as a Bayesian Approximation. So, in our approach all we need to do is keep sigma something as small as 10^(-34) ( for all practical purposes this is a zero for all discrete computers), the final objective function will become -\log(10^(-34)) + required terms = 34 + required terms ( a simplistic argument Hron provides a more rigorous one for the same). Hron et al. [3]  had called this approach the convolutional trick and quoting his work directly - “ we can view both the discretisation and convolutional approaches as mere alternative vehicles to derive the same quasi discrepancy measure”.  Hron et al. in their work “Variational Bayesian dropout: pitfalls and fixes” had never claimed standard Dropouts were irredeemably non-Bayesian.  They had also provided fixes to Gal’s original arguments by  the generalisation of variational inference to a quasi-KL and had shown ties between Gal’s convolution approach and their work. Thus, by extension, our method too can be considered Bayesian despite seemingly “infinite” KL divergence (Use Quasi KL divergence if uncomfortable with the convolution argument). Hron primary goal was to simply provide better-set arguments for Dropouts being Bayesian and these can be directly extended to our cases of ensembles being Bayesian with a mixture of Gaussian posterior.
>
> “The Bayesian bootstrap (Rubin, 1983) is a Bayesian interpretation of the bootstrap (identical under many conditions)”
> As pointed out correctly by you Ensembles are indeed very closely related to Rubin’s work Bayesian Bootstrap, but we will like to make further clarifications. Bayesian Bootstrap demands individual ensembles be trained on subsets of data, but this is not how ensembles are trained in practice. Our paper deals with the specific case of Ensembles mentioned in Lakshminarayanan et al. [1] where each ensemble is trained on the entire dataset, using randomized batch sampling to be data efficient. This “mathematically incorrect”  form (authors have claimed this to be a non-Bayesian approach) of training the ensemble has been empirically been shown give state of the art uncertainty estimates. Thus, we believe there is merit in our work in reframing this work into a Bayesian framework. We have also gone a step further and extended this to an ensemble of Bayesian NN’s like Dropouts, BBB etc. We have also added additional proof in the Appendix showing that our proof can be easily extended to an Ensemble of the arbitrary architecture of NN’s (making our proof model agnostic and proof far more general) which to the best of our knowledge are novel insights, and not well known in the literature. For example, just recently Ensemble of Dropout was proposed as a solution to the mode collapse issue in Active Learning by Remus et al.[ 2], but as pointed out by the reviewers on Openreview, they were unable to provide theoretical justification for this empirical finding.
>
> [1] Balaji Lakshminarayanan, Alexander Pritzel, Charles Blundell, NeurIPS 2017 Simple and Scalable Predictive Uncertainty Estimation using Deep Ensembles
> [2] Remus Pop and Patric Fulop - Deep Ensemble Bayesian Active Learning : Addressing the Mode Collapse issue in Monte Carlo dropout via Ensembles https://openreview.net/forum?id=Byx93sC9tm
> [3]Hron, Matthews & Ghahramani, Variational Bayesian dropout: pitfalls and fixes, ICML, 2018. https://arxiv.org/abs/1807.01969

---

> > ### Author Response · Authors · 2019-11-14
> > **Continuation**
> >
> >
> > “From a quick pass through Appendix A.2, the derivation is quite unstructured and difficult to follow. It is tough to see exactly what is being meant by the probabilities in the first line.
> > Additionally, the phrasing of “We further lower bound…” seems to suggest that the bound being used is not the well known bound of Durrieu et al, 2012, but a looser bound. If so, what is the approximation accuracy of this bound in relation to the other bounds compared against in this setting?”,
> >
> >
> > We apologise for the lack of clarity. Our goal was to show that when training an ensemble of Bayesian NN’s like BBB or Dropouts we are actually minimizing the variational free energy of a hypothetical Bayesian NN with mixture distribution. The lower bound derived by us is achieved by further lower bounding Durrieu et al. popular approximation to KL divergence between mixture distributions called the Variational Approximation. The lower bound is achieved when the pairwise KL divergence between individual components of  the mixture distribution is infinite (i.e. the individual components in the mixture are far apart). The key take away from this was that both the upper and lower bounds of the variational free energy with respect to the mixture distributions had the same form, summation of the variational free energy with respect to individual components in the mixture with only a difference of a constant entropy term $H(\frac{1}{N})$ (N is the number of components in the mixture). This implies that by training an ensemble of  BNN’s individually by minimizing their individual variational free energy we are training a hypothetical BNN with mixture distribution (sandwich theorem).
> >
> >  “ Furthermore, in Appendix 2, I see a potential issue in using the KL divergence and would like the authors to clarify.
> > If the secondary lower bound is being used to approximate the KL divergence between the mixture distributions, it becomes very unclear if what is being optimized is, in fact, a __lower bound__ on the log probability. After all, the standard ELBO is written as:
> > 	Log p(y) \geq ELBO := E_q(\log p(y | \theta)) – KL(q(\theta) || p(\theta))
> > Implying that another lower bound on the KL would be greater than the ELBO and thus not necessarily a lower bound on the log marginal likelihood.”
> >
> > We like to view training a BNN as minimizing the $KL(Q(W||\theta)||P(W||D))$ rather than the perspective of it being an upper bound of the -\log p(D). So all we are trying to do in the Appendix 2 is find an Upper and Lower Bound for $KL(Q(W||\theta)||P(W||D))$, where $Q(W||\theta)$ corresponds to a mixture distribution parameterized by $\theta_1,\theta_2, …,\theta_N$. The interesting nature of the derived bounds suggests that individual training of BNN by minimizing $KL(Q(W||\theta_i)||P(W||D))$ for i $\subseteq$ [1 to N] also minimizes the $KL(Q(W||\theta)||P(W||D))$, which would imply that we are also minimizing the \log p(y) too as $KL(Q(W||\theta)||P(W||D))$ is it’s upper bound.
> >
> > “The posterior is used only for fully connected layers…” Why is this the case? Alternative Bayesian methods can be used on the full network”
> >
> > Our method too can be used on the full network and as mentioned earlier in a much more general way (beyond MoG).  The specific setting we use is a simple trick for providing better uncertainty estimates than deep ensembles even with relatively less no. of parameters. The learnable dropout parameter (Concrete Dropout) on the convolutional layers has negligible cost and a full Bayesian Inference on only the final layer reduces the additional cost of our BNN to negligible. Thus at negligible parametric overhead can beat Ensembles which usually have a huge memory overhead, we believe this is particularly useful in resource-constrained applications. Also, this is the reason why we claimed our method to be far more scalable than standard Bayesian NN that rely on Mixture distributions.
> >
> >
> > “I am very concerned by the performance of the ResNet20 models trained on CIFAR10 in the paper”
> >
> > Due to computational constraints we had trained each model for 80 epochs only, and thus our accuracies are lower than the standard ones, we shall fix this issue in the updated version of the paper with an open-sourced code to verify the correctness of our implementation.
> >
> > We greatly appreciate the detailed reviews and regardless of the final decision would like your professional opinion on whether our work especially on ensembles being Bayesian (with the additional updates on further generalizing the proof to be model agnostic) is worth pursuing? We agree it seems like a low hanging fruit, but to the best of our limited knowledge, these insights are novel.

---

> > > ### Comment · AnonReviewer3 · 2019-11-15
> > > **Thank you for the clarifications**
> > >
> > > Performance issues: Thank you for pointing out that you were only able to train for 80 epochs. Until this issue is fixed completely, I still don't think that this paper can be accepted. If computational issues are still the concern, I'd look into advanced learning rate schedules (such as super-convergence https://arxiv.org/abs/1708.07120) to speed up the training procedure.
> > >
> > > Ensembles being Bayesian: In general, I think that yes, Bayesian interpretations of ensembles are potentially a fruitful avenue of research. However, in my personal (and potentially incorrect) view, I'd suggest looking back towards the rich statistics literature on the interface of the two methods rather than the modern machine learning literature.
> > >
> > > Bound: Thank you for the clarification - it's still a bit muddled and unclear. I'm not convinced that I fully buy the argument that if you alternately? optimize lower and upper bounds on the KL you will be provably optimizing a lower bound on the log evidence.

---

> > > > ### Author Response · Authors · 2019-11-15
> > > > **Further clarification**
> > > >
> > > > "it's still a bit muddled and unclear. I'm not convinced that I fully buy the argument that if you alternately? optimize lower and upper bounds on the KL you will be provably optimizing a lower bound on the log evidence"
> > > >
> > > > Understanding our proof is much easier if we view training a BNN as minimizing the KL divergence between an "assumed posterior" $q(w|\theta)$ and the true posterior, rather than minimization of the negative log evidence. Since in VI framework all we need are samples that seem to come from the true posterior.
> > > >
> > > > The upper and lower bounds are derived for the variational free energy w.r.t the mixture distribution (i.e. KL divergence between the mixture distribution and the "true posterior" ) and due to the interesting structure of the bounds training a single BNN i.e. minimizing the KL divergence between the individual component of mixture and the true posterior we will  be minimizing the Upper and Lower bounds simultaneously ( since both upper and lower bounds differ only by a constant entropy term).  Thus as the KL divergence between the mixture distribution and the true posterior is sandwiched between the upper and lower bound,  it too would be minimized. Thus we are training a "hypothetical" BNN with mixture distribution posterior by training an ensemble of BNN's.
> > > >
> > > > We understand your concern is that the optimal solution in both the cases, (training a mixture distribution BNN or an ensemble of BNN) will not be the same which is true and that there will be a bias. But due to the nature of the bound, the bias will decrease to zero as the size of the ensemble size grows to infinity ( since $\lim_{N\to\infty} H(\frac{1}{N}) = 0$ ). Even when dealing with finite-sized ensemble the bias will be bounded.
> > > >
> > > > "I'd suggest looking back towards the rich statistics literature on the interface of the two methods rather than the modern machine learning literature"
> > > >
> > > > Thank you for the suggestion we shall look into it.

---

> > > > > ### Comment · AnonReviewer3 · 2019-11-15
> > > > > **Response to clarification**
> > > > >
> > > > > "Since in VI framework all we need are samples that seem to come from the true posterior."
> > > > > This statement is somewhat incorrect - although it is the case that very broadly you only need to come up with an approximation distribution that is "close" to the true posterior, it's not clear if the solution to the optimization problem itself is reasonable in any way.
> > > > >
> > > > > "...rather than minimization of the negative log evidence" Instead, I believe you'd need to be minimizing KL(posterior || g) and would have to justify everything in that sense.
> > > > >
> > > > > "But due to the nature of the bound, the bias will decrease to zero ..." Would it be possible to show this numerically - e.g. via a set of Gaussians?

---

> > ### Comment · AnonReviewer3 · 2019-11-15
> > **Thank you for the further clarifications**
> >
> > It's interesting that you __can__ make the MC Dropout approximate posterior work by adding in the very tiny amount of noise. However, that's very much just a mathematical fix and not a real solution to the problem, which is that the optimized solution is a quasi-discrepancy instead.
> >
> > I'd encourage the authors to justify further why they would wish to be Bayesian in the first place. After all, ensembles themselves provide a relatively principled way of reasoning about uncertainty anyways. I think that justifying this would also make the paper stronger.

---

### Official Review · AnonReviewer1 · 2019-10-26
**Official Blind Review #1**

**Rating:** 3

**Review:**

The authors interpret Deep Ensemble as a special type of variational inference. Based on Bayes by Backprop which uses a Gaussian approximation to the posterior, the authors propose to use a mixture of Gaussians. The proposed methods have been tested on a regression task and Bayesian neural networks.

The authors argue that Deep ensemble is equivalent to variational inference with a mixture of Gaussians approximation with variance going to 0. It is not surprising that Deep ensemble is equivalent to variational inference with learning the mean only.  I’m not sure how useful this argument is since learning the distribution, not only the mean, is the key factor of being Bayesian.

Using a mixture of Gaussians rather than a Gaussian in Bayes by Backprop is a natural extension and therefore the novelty seems low.

In the experiments, I’m surprised to see that Bayes by Backprop fails to give any uncertainty estimate on the simple regression experiment. From the figure, BBB seems to be very confident about its prediction. However, it has been demonstrated in the literature that BBB is able to perform fairly well on this kind of regression. Can the authors give explanations on why it performs badly here? Also, I do not see why it is important to model multiple modes on this task. I believe a Gaussian approximation is able to work well. The multimodality here is likely induced by the symmetric parameterization of neural networks and trying to capture this kind of multimodality will be meaningless and even problematic. By looking at Figure 2, it seems like the posterior of the proposed method is close to being unimodal. Why is it beneficial to use a mixture of Gaussians under this situation?

The axis and labels in the figures are very small and hard to read.
Eq. (2) is overlapped with the above text.


**Experience Assessment:**

I have read many papers in this area.

**Review Assessment: Checking Correctness Of Derivations And Theory:**

I assessed the sensibility of the derivations and theory.

**Review Assessment: Checking Correctness Of Experiments:**

I carefully checked the experiments.

**Review Assessment: Thoroughness In Paper Reading:**

I read the paper at least twice and used my best judgement in assessing the paper.

---

> ### Author Response · Authors · 2019-11-11
> **Thank you for the feedback**
>
> We thank reviewer for his comments. We try to address his concerns in the best way possible.
>
> “It is not surprising that Deep ensemble is equivalent to variational inference with learning the mean only.  I’m not sure how useful this argument is since learning the distribution, not only the mean, is the key factor of being Bayesian.”.
>
> Deep Ensembles are well known to have ties with Bayesian Inference since they can be easily related to “The Bayesian Bootstrap” by Rubin [1], where individual ensembles are trained on subsets of data, but this is not how it is used in practice.  Lakshminarayanan et al. in their work “Simple and Scalable Uncertainty Estimation Using Deep Ensembles” [2] empirically showed that by using a mathematically incorrect form of training all ensembles on the same data, albeit using a randomized batch sampling, we can still obtain state of the art uncertainty estimates. This approach is massively popular for all practical purposes since it has much higher data efficiency, which was the main novelty of their work. The importance of our work comes from the fact that we are able to show that ensembles relying on randomized batch sampling can be viewed as Variational Inference with “Mixture of Gaussian Posterior” (Not impulses, to avoid pathological KL divergences). We were also able to extend this viewpoint to an Ensemble of Bayesian Neural Networks like an ensemble of Dropout or BBB’s, which to the best of our knowledge is also novel. Past works have shown ensembles of Dropouts have been shown to be able to be more robust to adversarial perturbations and be able to solve the issue of mode collapse in Active Learning Scenarios [3], but they were unable to provide theoretical justifications for their empirical findings, thus we would argue that our work is useful in the sense it provides theoretical support and is consistent with past works.
> Another reason why our view is of importance is because our proof can be easily extended and made model agnostic. In the sense that by using a small trick, we can show that an ensemble of arbitrary architecture is also Bayesian which to the best of our knowledge is not known. We have added this simple extension in the Appendix, view Appendix 3 for details.
>
>
>
> “key factor of being Bayesian”
>
> We would argue that a key factor to being Bayesian is being able to get samples which seem to be derived from the true posterior, which ensembles are fully capable of and thus are Bayesian in Nature. Do note that  Ensembles aren’t the only particle filter like approach to approximating the True Posterior, Stein Variational Gradient Descent [4]  mentioned in our paper are similar to Ensembles and are known to be Bayesian.
>
>
>
> “Using a mixture of Gaussians rather than a Gaussian in Bayes by Backprop is a natural extension and therefore the novelty seems low.”
>
> Indeed our work can be seen as a natural extension of BBB, but the novelty lies in the fact that using our method, we can design arbitrary complex concrete mixture distributions beyond the Gaussian ones used in the paper, e.g. our method can be combined with Normalizing Flows to model fairly complex distributions. We have also proposed and shown a fairly novel way to train Bayesian Neural Networks by treating the Convolution Layers as some type of dimensionality reduction algorithm (learnable dropouts using concrete distribution used here ) and performing  Bayesian inference on only the final fully connected layer. This trick will be of exceptional importance for scaling to large scale applications or in memory constraint applications.
>
> “Why multimodality”?
> As observed in our Fig 2, indeed two modes are very close but still unimodal distributions would not have captured the third distinct mode as in this case. The multimodality advantage will only increase as moving onto more complex regression or other classification tasks.
> We shall address your remaining concerns regarding the experiments ( of BBB) soon after open-sourcing the code.
>
>
>
> [1] Rubin, The Bayesian Bootstrap, 1981. Annals of Statistics.
> https://projecteuclid.org/euclid.aos/1176345338
> [2] Balaji Lakshminarayanan, Alexander Pritzel, Charles Blundell, NeurIPS 2017. Simple and Scalable Predictive Uncertainty Estimation using Deep Ensembles
> [3]Remus Pop and Patric Fulop - Deep Ensemble Bayesian Active Learning : Addressing the Mode Collapse issue in Monte Carlo dropout via Ensembles https://openreview.net/forum?id=Byx93sC9tm
> [4] Qiang Liu and Dilin Wang. Stein variational gradient descent: A general-purpose Bayesian inference algorithm. In Advances in neural information processing systems

---

### Official Review · AnonReviewer4 · 2019-11-04
**Official Blind Review #4**

**Rating:** 3

**Review:**

Summary
In this works, the authors propose to use a concrete mixture of Gaussians as a variational distribution. The authors show that the deep ensemble method can be viewed as a special case of a mixture of Gaussians with a categorical prior.
I think the main contribution is to use the concrete distribution as a mixture prior q(z) instead of a categorical prior.
However, there are some concerns. The following points should be addressed to get a higher rating.

(1) Several papers consider the problem of learning a mixture of Gaussians in the VI framework ([1,2,3,4,5]). The deep ensemble method considered in this paper is just one of the existing ensemble approaches. The related work section should be updated to discuss the novelty of this work.

(2) In the deep ensemble method, the categorial prior q(z) is held fixed. However, it is possible to update the categorical prior q(z).
In this work, the proposed distribution is q(w) = sum_{c=1}^{K} Concrete(z=c|\theta_z) Gauss(w|z=c,\theta_{w_c}).
By using the concrete prior, the gradient can be computed by the local reparametrization trick for q(w|z) and the reparametrization trick for q(z).
However, even when q(z) is the categorical prior,  the local reparametrization trick for q(w|z) is still valid if the variational parameters for each mixture component are not tied.  The main difference is the reparametrization trick can not be used for q(z).
The authors should show why the proposed variational distribution is better than the mixture of Gaussians with a categorical prior.


References
[1] O. Arenz, M. Zhong, and G. Neumann. "Efficient Gradient-Free Variational Inference using Policy Search." ICML. 2018.
[2] A. C. Miller, N. J. Foti, A. D’Amour, and R. P. Adams. Variational boosting: Iteratively refining posterior approximations. ICML, 2017.
[3] O. Zobay. Variational Bayesian inference with gaussian-mixture approximations. Electronic Journal of Statistics, 8(1):355–389, 2014.
[4] Lin, Wu, Mohammad Emtiyaz Khan, and Mark Schmidt. Fast and Simple Natural-Gradient Variational Inference with Mixture of Exponential-family Approximations. ICML 2019.
[5] F. Guo, X. Wang, K. Fan, T. Broderick, and D. B. Dunson. Boosting variational inference. arXiv:1611.05559v2, 2016.





**Experience Assessment:**

I have published one or two papers in this area.

**Review Assessment: Checking Correctness Of Derivations And Theory:**

I assessed the sensibility of the derivations and theory.

**Review Assessment: Checking Correctness Of Experiments:**

I assessed the sensibility of the experiments.

**Review Assessment: Thoroughness In Paper Reading:**

I read the paper at least twice and used my best judgement in assessing the paper.

---

> ### Author Response · Authors · 2019-11-14
> **Thank You for the Pointers**
>
> “I think the main contribution is to use the concrete distribution as a mixture prior q(z) instead of a categorical prior”, “Several papers consider the problem of learning a mixture of Gaussians in the VI framework ([1,2,3,4,5])”
>
> Our key goal was to make Bayesian Neural Networks (BNN) easily to train and scalable to large scale Neural Networks so another key novelty in our method is using Learnable Dropouts in the Convolutional Network and performing full Bayesian Inference with mixture distribution on only the final layer. This makes our approach several times less costly in terms of memory,  while still achieving state of the art uncertainty estimates on out of distribution (OOD). So our work is different from the ones cited by you in the sense we want to design a practical BNN which can replace standard Neural Networks for a limited overhead (besides the MC sampling the complexity is almost the same as standard NN, making this a simple but effective trick).
>
> “The authors should show why the proposed variational distribution is better than the mixture of Gaussians with a categorical prior“
> We agree a comparison between the two is due and we plan to add a comparison of NLL and Expected Calibration Error comparison for the same in the updated version of the paper, with the open sourced code. An easy intuition as to why our method is guaranteed to be better is that by tweaking the training procedure such that we keep the parameter p’s of the categorical random variable fixed until convergence and then switch a flag to make p’s tunable will guarantee a lower ELBO for our case.
>
> “The deep ensemble method considered in this paper is just one of the existing ensemble approaches”
>
> Indeed there are multiple ways to the ensemble and we are referring to the specific case described by Lakshminarayanan et al. [1]. Links between Ensembles and Bayesian are neither new nor groundbreaking as it is well known that Ensembles are closely linked to the Rubin’s Bayesian Bootstrap [2]. Rubin’s method demands each ensemble be trained on an independently sampled subset of data but this is not how ensembles are trained in practice. In order to be data efficient we usually train each ensemble on the entire data albeit using a randomized mini-batch sampler. This “mathematically incorrect” form of training causes it to lose the theoretical support that other BNN’s enjoy. We show that this form of “randomized mini-batch” training can be reframed into the VI framework. A merit of this viewpoint is that we can easily generalize our proof to an ensemble of Neural Networks with arbitrary architectures (updated in the Appendix) which is completely novel. We have also shown that ensemble of BNN’s like Dropouts or BBB too can be considered Bayesian. Just recently Remus  et al. had [3] proposed ensemble of Dropouts as a solution to mode collapse issue in Active Learning but as pointed out by reviewers on OpenReview they were unable to provide theoretical justification for their empirical findings.
>
>
> [1] Balaji Lakshminarayanan, Alexander Pritzel, Charles Blundell, NeurIPS 2017 Simple and Scalable Predictive Uncertainty Estimation using Deep Ensembles
>
> [2] Rubin, The Bayesian Bootstrap, 1981. Annals of Statistics. https://projecteuclid.org/euclid.aos/1176345338
>
> [3] Remus Pop and Patric Fulop - Deep Ensemble Bayesian Active Learning : Addressing the Mode Collapse issue in Monte Carlo dropout via Ensembles https://openreview.net/forum?id=Byx93sC9tm

---

> > ### Comment · AnonReviewer4 · 2019-11-15
> > **Request for further clarifications**
> >
> > " another key novelty in our method is using Learnable Dropouts in the Convolutional Network and performing full Bayesian Inference with mixture distribution on only the final layer"
> >
> > If my understanding is correct, p(w_{-K}, w_K) is parameterized by a NN, where w_K is the weight of the final layer. The authors would like to use a variational distribution q(w_K) such as a Gaussian mixture to perform inference. Is my understanding correct? If this is the case, we can use a variational distribution q(w)=q(w_{-K})q(w_K) where q(w_{-K}) is a dropout distribution and q(w_K) is a Gaussian mixture with a categorical prior. This variational distribution is also "easy to train and scalable to large scale Neural Networks." Furthermore, we also learn "Dropouts in the Convolutional Network" using q(w_{-K}), and perform "Bayesian Inference with mixture distribution on only the final layer"using q(w_K).
> >
> >
> > Bottom line: You should show a concrete mixture distribution is better than a categorical mixture distribution. For example, compared to the categorical mixture, the concrete mixture distribution is faster to train while it gives similar performance. You should tell readers the advantages and disadvantages of using the concrete mixture distribution.

---

> > > ### Author Response · Authors · 2019-11-15
> > > **Clarifications**
> > >
> > > Indeed our method is a combination of Convolutional Concrete Dropout (for pre-final layers) and Mixture Distribution Variational Inference (for final layers). And the factorization mentioned by you is indeed correct. Thus making it both lightweight and flexible, allowing it to outperform Deep Ensembles that have approximately 3 times its space complexity.
> > >
> > >
> > > "You should show a concrete mixture distribution is better than a categorical mixture distribution. For example, compared to the categorical mixture"
> > > True a comparison is necessary, we will be adding a comparison of their expected calibration error to the uploaded version of the paper, along with a discussion of the cost.
> > >
> > > Another advantage of our method is in Concrete Ensembles, specifically in the case when the architectures of the NN's in the ensemble are not the same. When dealing with Ensembles of NN's like Alexnet and Resnet (architecture of the ensemble members may be different) we know Resnet's are superior and deserve more weight. But a categorical mixture distribution will give equal weight to each member of the ensemble, and here using concrete distribution to learn the importance of each member will become an absolute must.
> > >
> > > We are aware there are other alternatives to finding the importance of  NN's in an ensemble the most popular being their weighted accuracy on the validation set, but our method of using a concrete mixture distribution provides a principled Bayesian approach for the same.

---

> > > > ### Comment · AnonReviewer4 · 2019-11-15
> > > > **Thank you for the clarifications**
> > > >
> > > > Please correct me. Otherwise, I assume q(w_{-K})q(w_K) is the combination of  "Convolutional Concrete Dropout and Mixture Distribution". If this is the case, I think this novelty is not significant since it is a straightforward combination.
> > > >
> > > >
> > > > " a categorical mixture distribution will give equal weight"
> > > > Please note that q(w) = \int q(z) q(w|z) dz
> > > > I assume that q(z) is held fixed when q(z) is a categorical distribution.
> > > > As mentioned in my initial review,  you can *jointly update* the mixing weight q(z) and each component q(w|z) even when q(z) is a categorical distribution instead of a concrete distribution. In the references cited in my initial review, q(z) can be learned even when q(z) is a categorical distribution.
> > > > I still don't think that this paper can be accepted unless you show that a concrete mixture distribution is better than a mixture distribution with a learned categorical mixture prior.

---

> > > > > ### Author Response · Authors · 2019-11-15
> > > > > **Do consider ensembles as Bayesian a part of the novelty in our work**
> > > > >
> > > > > "If this is the case, I think this novelty is not significant since it is a straightforward combination."
> > > > >
> > > > > Do consider our work in linking Deep Ensembles to the Variational Framework as a part of the novelty. We have also successfully extended our proof to an Ensemble of NN's with arbitrary architectures using a slight trick, view Appendix 3 for details, which to the best of our knowledge is novel. We have also argued in Section 3 of our papers ensembles of BBB, dropout too can be seen as a Bayesian approximation which as per our citation [3] is not well known either.
> > > > >
> > > > >
> > > > > [3] Remus Pop and Patric Fulop - Deep Ensemble Bayesian Active Learning: Addressing the Mode Collapse issue in Monte Carlo dropout via Ensembles https://openreview.net/forum?id=Byx93sC9tm

---

### Author Response · Authors · 2019-11-15
**General Comments for all reviewers**

We have extended the proof of Ensembles as described in  [1]  being bayesian to a more general setting where the architectures of the members in the  Ensembles need not be the same, view Appendix 3 for the details.

We have open-sourced the code for two experiments, the uncertainty plots of  MoG in case of adversarial attack one, and MoG Reset on CIFAR. We have updated the results of the CIFAR experiment after running each network for 200 epoch and also added the Expected Calibration Error comparison of CMoG Bayesian NN, MoG Bayesian NN (keeping the p's fixed in MoG) and Deep Ensemble.




[1] Balaji Lakshminarayanan, Alexander Pritzel, Charles Blundell, NeurIPS 2017 Simple and Scalable Predictive Uncertainty Estimation using Deep Ensembles

---

### Decision · Program_Chairs · 2019-12-19

**Decision:**

Reject

**Comment:**

This paper proposes to use mixture distributions to improve uncertainty estimates in BNNs. Ensemble methods are interpreted as a Bayesian mixture posterior approximation. To reduce the computation, a modification to BBB is provided based on a concrete mixture distribution.

Both R1 and R3 have given useful feedback. It is clear that interpretation of ensemble as a Bayesian posterior is well known, and some of them also have theoretical issues. The experiment to clearly comparing proposed mixture posterior to more commonly used mixture distribution is also necessary.

Due to these reasons, I recommend to reject this paper. I encourage the authors to use reviewers feedback to improve the paper.